# Development and Evaluation of an Axiom^TM^ 60K SNP Array for Almond (*Prunus dulcis*)

**DOI:** 10.3390/plants12020242

**Published:** 2023-01-05

**Authors:** Henri Duval, Eva Coindre, Sebastian E. Ramos-Onsins, Konstantinos G. Alexiou, Maria J. Rubio-Cabetas, Pedro J. Martínez-García, Michelle Wirthensohn, Amit Dhingra, Anna Samarina, Pere Arús

**Affiliations:** 1Unité de Génétique et Amélioration des Fruits et Légumes (GAFL), INRAE (French National Research Institute for Agriculture, Food and Environment), 84143 Montfavet, France; 2Centre for Research in Agricultural Genomics (CRAG) CSIC-IRTA-UAB-UB, Carrer de la Vall Moronta, Edifici CRAG, Campus UAB, Cerdanyola del Valles, 08193 Barcelona, Spain; 3IRTA (Institute of Agrifood Research and Technology), Campus UAB, Edifici CRAG, Cerdanyola del Valles (Bellaterra), 08193 Barcelona, Spain; 4CITA (Agrifood Research and Technology Centre of Aragon), Department of Plant Science, Avda. Montañana 930, 50059 Zaragoza, Spain; 5CEBAS (Centro de Edafología y Biología Aplicada del Segura), CSIC, Department of Plant Breeding, Campus Universitario de Espinardo, 30100 Espinardo, Spain; 6Waite Research Institute, University of Adelaide, PMB 1 Glen, Osmond, SA 5064, Australia; 7Department of Horticulture, Washington State University, Pullman, WA 99164-6414, USA; 8Thermo Fisher Scientific, Frankfurter Str. 129B, 64293 Darmstadt, Germany

**Keywords:** *Prunus dulcis*, *Prunus persica*, SNP, Axiom array, genotyping

## Abstract

A high-density single nucleotide polymorphism (SNP) array is essential to enable faster progress in plant breeding for new cultivar development. In this regard, we have developed an Axiom 60K almond SNP array by resequencing 81 almond accessions. For the validation of the array, a set of 210 accessions were genotyped and 82.8% of the SNPs were classified in the best recommended SNPs. The rate of missing data was between 0.4% and 2.7% for the almond accessions and less than 15.5% for the few peach and wild accessions, suggesting that this array can be used for peach and interspecific peach × almond genetic studies. The values of the two SNPs linked to the *RMja* (nematode resistance) and *SK* (bitterness) genes were consistent. We also genotyped 49 hybrids from an almond F2 progeny and could build a genetic map with a set of 1159 SNPs. Error rates, less than 1%, were evaluated by comparing replicates and by detection of departures from Mendelian inheritance in the F2 progeny. This almond array is commercially available and should be a cost-effective genotyping tool useful in the search for new genes and quantitative traits loci (QTL) involved in the control of agronomic traits.

## 1. Introduction

The detection of single nucleotide markers (SNPs) has been enhanced with the Next-Generation Sequencing (NGS) technologies [1,2]. SNPs used as molecular markers in linkage mapping studies and genome wide association studies (GWAS) have enabled the detection of novel genes or quantitative trait loci (QTL). The SNP array technologies have facilitated the SNP genotyping by providing complete and reliable genotypic data at an acceptable cost [3].

Several high density SNP arrays have been developed in fruit species, either with the Illumina technology in the 9K [4] and 18K SNP peach array [5] and the 15K cherry SNP array [6], or with the Axiom Affymetrix technology in the 480K apple SNP array [7], the 70K and 200K Pyrus SNP array [8,9] or the 700K SNP walnut array [10]. 

Based on the strong synteny between peach (*Prunus persica*) and almond (*Prunus dulcis*), the Illumina 18K peach SNP array was used in a GWAS analysis of a collection of Italian almond germplasm [11]. The authors obtained an average marker density of 1 SNP per 424 kb across the almond genome. With the development of an almond-specific array, this should improve the marker density and cover the whole almond genome, in particular its almond-specific regions.

The whole genome sequences of *Prunus dulcis* cv “Texas” [12] and *Prunus dulcis* cv “Lauranne” [13] were published and are available on the GDR database (https://www.rosaceae.org/analysis/295 accessed on 1 January 2021), and recently another whole genome cv “Nonpareil” was also published [14]. In addition to the sequencing of the “Texas” genome, the “almond genome” consortium has resequenced several almond cultivars using paired-end Illumina sequencing, allowing the detection of a high number of SNPs and InDel variants.

In this study, we present the development of an Axiom 60K almond SNP array (PdSNP_V1) based on the SNP polymorphism of a set of 81 resequenced almond accessions and the validation and accuracy of the genotyping data generated from the array. 

## 2. Results–Discussion

### 2.1. Array Design

#### 2.1.1. SNPs Array Selection 

Alignment of the NGS reads of the 81 almond accessions to the “Texas” reference genome [12] and SNP calling with GATK (Section 3.2) allowed us to detect 11,683,655 SNPs. We then applied three different steps (Figure 1), to select SNPs for the chip. In the first step with the filters described in Section 3.2, we obtained 576,968 SNPs with the threshold of less than 20% missing data (NA) and 71,843 SNPs for less than 10% and we retained the 10% NA filter. We did not discard the A/T and C/G SNPs although they require two probes per SNP in the Axiom probe sets. In the second step, we included the three additional SNPs published as trait markers of interest and detailed in Section 3.2 and Table 1.

The final filter on step 3 was based on the p-convert score estimated by Affymetrix. Prior to chip production, an in-silico validation was performed to estimate the p-convert value for the forward and reverse probes for each SNP. The p-convert value is derived from a random forest model to predict the probability of SNP conversion on the array. A high p-convert value means that we expect a high probability of conversion for the SNP probes on the array. SNPs with a p-convert of less than 0.666 were removed, discarding 11,265 SNPs. In addition, Affymetrix had classified SNPs into three categories, recommended, not recommended and neutral (Figure 2). Only SNPs classified as recommended were retained for chip production. In the final selection, 60,581 SNPs that met all the criteria were used to construct the almond chip.

#### 2.1.2. Distribution of Array SNPs

To explore the distribution of the 60,581 selected SNPs, SNP frequency was plotted at every one million base pairs along each chromosome. (Figure 3). We have also plotted the distribution of all 576,968 SNPs detected after step 1 of the pipeline selection. The density of SNPs seems highest along the chromosome arms and lowest in the centromeric regions for both distributions. Over these million-base-pair intervals, the number of SNPs on the chip ranges from 22 to 799 (Appendix A) and there are no SNP-free intervals. The average distance between two SNPs is 2891 bp. 

### 2.2. Validation of the Axiom 60K Almond Array

#### 2.2.1. Validation of the Array on the Diversity Panel 

To evaluate and validate the Axiom 60K almond SNP array, we genotyped a diversity panel composed of 210 accessions including 187 genotypes of *P. dulcis* and 23 genotypes of species of the subgenus *Amygdalus* including some peach genotypes and interspecific hybrids. A total of 47,012 SNPs (77.6%) were classified in the Poly High Resolution (PHR) Affymetrix category, considered the most informative with good cluster resolution (Table 2). Only 0.1% (85 SNPs) were classified in the “Mono High Resolution” (MHR) category, which means that all samples are monomorphic for these SNPs. A total of 5.1% were classified in the “No Minor Homozygous” (NMH) category which includes SNPs with good cluster resolution, but no minor homozygous genotypes. The other categories “Call Rate Below Threshold” (CBT), “Off-Target Variant” (OTV), Other (low quality), not recommended for downstream analysis, accounted for 17.2% of SNPs. 

For all 60,581 SNPs, we observed between 262 (0.4%) and 1665 (2.7%) missing data for the 185 *P. dulcis* accessions (Table 3). In comparison, in the data generated by NGS resequencing of the 55 almond accessions resequenced by the consortium, we obtained between 1.6% and 7.6% missing data, showing a higher rate of missing data than in the chip data. For the wild almond species (*P. bucharica, P. fenzliana, P. kuramica, P. dehiscens, P. webbii*), the array data are also valuable with a similar frequency of missing data as *P. dulcis*, but with a lower percentage of heterozygous SNPs. For accessions of the two peach species, *P. persica and P. davidiana*, the rate of missing data was slightly higher (11.7–15.5%), but it is still a low rate. With a relatively high number of heterozygous SNPs, this suggests that the array has the potential to be used for diversity analysis and linkage mapping of peach trees, and specifically for linkage mapping of peach x almond progenies, useful for gene transfers from almond to peach or vice versa [15].

#### 2.2.2. Validation of the Three Additional SNPs

In the almond diversity panel analysis, the three additional SNPs were classified in the category Poly High Resolution (PHR) and were considered “best recommended” (Appendix A). For the SNP marker AX-599403222 of the *RMja* gene [16], only the two resistant accessions R1107 (Alnem-1) and R1109 (Alnem-88) were genotyped Homozygote with the alternative allele, which was consistent with their resistant phenotypes [17]. Three wild accessions and seven almond accessions were genotyped heterozygote for this SNP marker. As the *RMja* gene is a dominant gene, these genotypes would be resistant, which will be validated by an inoculation test with root-knot nematodes. All other almond accessions were genotyped homozygous susceptible. This is consistent with published data [18] and results obtained in resistance tests of four almond accessions (Ferragnès, Ferrastar, Dorée and Pointue d’Aureille) against *M. javanica* (data not published). The *RMja* gene is an orthologue of the plum gene *Ma* [19] and the genotyping with the almond chip of the three resistant clones P1079 (*MA/MA*), P2175 (*Ma/ma*) and P2980 (*Ma/ma*) was well determined by the SNP AX-599403222.

For the SNP AX-599403226, including in the *Prudu_000307-V1.0* gene, which could be associated with kernel weight [20], we found a segregation AA/AB/BB of 13/45/152 in the panel (Appendix A). With our kernel weight data available on some almond accessions, we did not observe statistically significant differences between the three classes, so we could not confirm the association. However, this SNP is classed in the Poly High Resolution class h justifying it to be a good SNP marker.

For the SNP AX-599403227 located in the region of the *Sk* gene involved in the bitter taste of almond [13], we found segregation AA/AB/BB of 70/120/20 for the panel of 217 accessions. The majority of almond varieties are sweet according the genotyping score (AA or AB), confirming previous studies. As previously observed by Sanchez et al. [21], the percentage of bitter phenotypes was very low, as a result of the selection against the recessive allele of the sweet kernel (*Sk*) in the breeding program. Twenty accessions were scored BB (Appendix A) and amongst them, there were the five bitter accessions of the panel, but also 15 sweet accessions. This discrepancy in “Atocha” was observed previously by Sánchez-Pérez et al. [13]; a different SNP in the same candidate gene (*bHLH2*) could be the main reason for the sweet phenotype according these authors. In the case of “Texas” and the other sweet accessions that scored ‘BB’, the observed discrepancy could be explained by a different origin of the sweet phenotype, as result of a different SNP mutation in the same or other unknown gene; this still remains unclear. In conclusion, this marker can confirm that with the genotype AA or AB, the variety is sweet, but with the genotype BB we could not confirm it is bitter. However, this SNP has a good profile and high resolution, and a good marker of this region (Appendix A).

#### 2.2.3. Validation of the Array with a F2 Progeny

To construct a genetic map for the F2 almond progeny, we first selected heterozygous SNPs of the F1 parent “Penta” from the files generated by the almond chip and analyzed with “Axiom Analyze Suite”. These SNPs were classified in the two categories Poly High Resolution (PHR) (10,624 SNPs) and No Minor Hom (50 SNPs). We retained only 7234 intercross SNPs with a co-dominant 1:2:1 segregation, with a p-value less than or equal to 0.01 for a chi-squared goodness-of-fit test [22]. Following the hypothesis of no dominance between alleles, we coded these markers as *hkxhk,* as in a cross-pollination (CP) in JoinMap^®^4.1 [23]. For grouping, an LOD ≥ 4 was used. The Maximum Likelihood (ML) mapping algorithm and the Kosambi mapping function implemented in JoinMap^®^ were applied to order and determine the genetic location of markers. In order to keep the most reliable markers, we chose to remove markers mapping far from any other marker with a threshold of 5 cM (centimorgan). Finally, the selection and the construction of the genetic map were performed in JoinMap^®^4.1, resulting in a final set of 1159 SNPs (Table 4). The map size was 876 cM and the mean distance between two SNP markers was 0.76 cM.

Our results show that the almond chip PdSNPV1 can be useful for constructing high-density linkage maps for almond and peach × almond populations. Some linkage maps with SNP markers have already been constructed in almond and peach × almond populations using the SNP Illumina peach chip [24] or genotyping by sequencing (GBS) technology [25,26], but this almond chip could generate more specific almond SNP markers with greater ease and lower cost.

### 2.3. Evaluation of the Error Rates in Different Analyses 

#### 2.3.1. Comparison of “Texas” Genotyping with the “Texas” Reference Genome

The genotypic array data from the “Texas” accession was compared to the data of the “Texas” reference genome. Table 5 represents the number of SNPs that belong to one of the three allelic states 0, 1 or 2 with “0” for no common allele with the reference genome (alternative homozygote), “1” for only one common allele with the reference genome (heterozygote), “2” for two common alleles with the reference genome (reference homozygote). With the array data of “Texas”, we found 109 SNPs as alternative homozygotes to the reference genome. They are considered genotyping errors, meaning that the error rate was around 0.2% in the “Texas” array data. This could be also considered clonal variation, between the two different “Texas” clones used in the DNA extraction. The number of heterozygotes (10,416) is high but within the range observed in the interval observed in the 185 almond accessions of Table 3. 

#### 2.3.2. Comparison of Array Data from Three Replicates of the “Ferrastar” Variety

To assess the repeatability of the genotyping, we tested three “Ferrastar” samples. The differences of allelic state between each replicate of Ferrastar were the consequence of the natural error rate belonging to the Axiom technology. There were 0.90% and 0.95% error rates between the first replicate and the two other replicates and 1.01% between the second and the third replicate (Table 6), which is considered a low error rate.

#### 2.3.3. Comparison between Expected and Observed Homozygote F2 Data 

For all homozygous SNP types detected in the F1 parent “Penta”, the 49 F2 hybrids were expected to have the same allelic status as the parent. With 20,462 monomorphic markers AA in Penta for the A allele of “Texas”, we expected only AA datapoints for these 20,462 markers in the 49 F2 hybrids, but 494 were genotyped AB and 2167 were BB on the 1,002,203 observed data points corresponding to 0.27% erroneous data (Table 7). For the alternative B allele of “Texas”, there were 22,276 monomorphic markers BB in Penta, and in the 49 hybrids, 1324 data points were AA and 332 were AB on the 1,091,205 data points corresponding to 0.15% erroneous data The error rate was lower than in the other two previous comparisons and also lower than the results presented in the 70k SNP Pyrus chip [8] where there was an error rate of 3%.

## 3. Materials and Methods

### 3.1. Plant Material Resequenced

We resequenced 50 accessions of *P dulcis* and 5 of wild species, two of *P. webbii* and one of *P. bucharica*, *P. kuramica*, and an interspecific hybrid *P. fenzliana* × *P. bucharica.* Forty-two accessions were resequenced on Illumina^®^ HiSeq2000 by the CNAG-CRG Barcelona (https://www.cnag.crg.eu/ accessed on 1 June 2020) and the 14 others were resequenced on Novaseq 6000 Illumina at the MGX platform at Montpellier (https://www.mgx.cnrs.fr/ accessed on 1 January 2021)). For increasing the panel, we downloaded the sequences available on NCBI, 14 published by Yu et al. [27], 3 by Koepke et al. [28], and 9 by Velasco et al. [29] (Appendix A). This panel of 81 accessions included a wide genetic diversity encompassing American, Australian, French, Italian, Spanish and Russian origins (Figure 4) and the three cultivars: “Tuono”, “Cristomorto” and “Nonpareil” which are the main genitors used in almond breeding programs [30].

All fastq files were trimmed using Trim Galore! version 0.6.1 [31] and their quality control was checked before and after trimming using FastQC v0.11.5 [32]. The trimmed fastq files for the 81 resequenced accessions were aligned to the “Texas” almond reference genome v2.0 [12] (https://www.rosaceae.org/ accessed on 1 January 2021) using BWA version 0.7.16a-r1181 [33]. Samtools (v1.9) [34] was used to convert SAM into BAM format, sort and index the BAM files.

### 3.2. SNP Calling and Selection of SNPs for Array Development

All fastq files were trimmed using Trim Galore! version 0.6.1 [31] and their quality control was checked before and after trimming using FastQC v0.11.5 [32]. The trimmed fastq files for the 81 resequenced accessions were aligned to the “Texas” almond reference genome v2.0 [12] (https://www.rosaceae.org/ accessed on 1 January 2021) using BWA version 0.7.16a-r1181 [33]. We used GATK v4.1.3 [35] to perform the variant calling with the following VariantFiltration expression: QD (QualityByDepth) < 2.0, SOR (StrandOddsRatio) > 3.0, FS (FisherStrand) > 60.0, MQ(RMSMappingQuality) < 40.0, MQRankSum < −12.5, ReadPosRankSum < −8.0, ExcessHet > 54.69, DP (Depth) > 10.

After this initial filtering, we applied three more selection steps (Figure 1). In the first step, SNP selection was done for biallelic SNPs, absence of other SNPs around 30 nucleotides, not overlapping with a transposable element, no missing data >10% and a Minor Allele Frequency (MAF) higher than 5%. In the second step, we included 3 additional SNPs published as markers for interesting traits (Table 1). The SNP AX-599403222 is the SNP of the KASP marker SP903 designed in the root knot nematode resistance *RMja* gene [16]. The AX-599403226 SNP is a SNP detected in the region of the mutation of the bHLH (basic helix-loop-helix) transcription factor (gene *Sk*) involved in the bitter taste of almond [13]. The last SNP, AX-599403226 located on the group 1, was detected by a GWAS signal for kernel weight [36]. We were not aware of any other SNP markers linked to other agronomic traits in almond and only these 3 SNPs were included in the chip. In the last selection step, the selected SNPs and corresponding flanking sequences were submitted to Affymetrix for initial probe screening and to estimate a P-convert score for each SNP.

### 3.3. Plant Material for the 60K SNP Array Validation

To evaluate and validate the Axiom 60K SNP almond array, a diversity panel was composed including 210 accessions made up of 187 of *P. dulcis* from the INRAE and IRTA collections, and other accessions of species of the subgenus *Amygdalus* (Table 3). Three replicates of the accession “Ferrastar” and the reference “Texas” were included in this panel. We also used an F2 progeny of 49 individuals from a self-pollination of the self-fertile Spanish accessions “Penta” issued from the cross “Lauranne” × S5133 [37], and considered as F1 hybrid.

### 3.4. Genotyping and SNP Analysis

Genomic DNA was extracted from young leaves according to the protocol of Antanaviciute et al. [38]. The amount of DNA required for genotyping with the array was 50 µL/genotype at 10 ng/µL. Genotyping was performed on an Axiom GeneTitan system on the INRAE Gentyane platform at Clermont-Ferrand (https://gentyane.clermont.inrae.fr/ accessed on 1 November 2021).

After genotyping accessions, genotype calling and QC metrics were performed with SNP Axiom Analysis Suite v5.1 (Affymetrix) using diploid threshold configurations and default DishQC settings (DQC ≥ 0.82 and call rate > 0.97).

The software classified SNPs into six categories: Poly High Resolution (PHR) when we observed a good resolution for the two homozygous clusters and at least two occurrences of the minor allele, Mono High Resolution (MHR) when the SNP passes all thresholds except the number of minor alleles., meaning that all genotyped samples are monomorphic, No Minor Hom (NMH) when one of the homozygote clusters was not observed; Off-Target Variant (OTV) when the SNP has a sequence significantly different from the sequence of the hybridization probes caused for example by double deletions or sequence non-homology, Call Rate Below Threshold (CBT) when the SNP call rate is below the ratio of 97, meaning that the software has failed to identify clusters, “Other” when the SNP does not satisfy all quality thresholds.

## 4. Conclusions

We have developed and validated a new Axiom 60K almond SNP array (PdSNP_V1) and our studies have shown that the genotyping is reliable, with few missing data and erroneous points. Based on F2 progeny, we found only a few errors indicating departures from Mendelian inheritance of homozygote SNP loci.

The PdSNP_V1 array could be used for almond diversity analysis, association studies and genetic mapping for breeding purposes, as a large number of breeding genitors were included in the set of accessions resequenced for the detection of SNPs. Our data showed that this array is also effective for peach species and some wild *Amygdalus* species. The PdSNP_V1 array would be suitable for *Prunus* rootstock breeding, notably for linkage mapping of peach × almond progenies. The PdSNP_V1 array is the first developed almond array and is commercially available from Thermofischer Scientific. SNP genotyping can be performed on INRAE’s Gentyane platform or on other platforms equipped with Axiom genotyping instruments.

## Figures and Tables

**Figure 1 plants-12-00242-f001:**
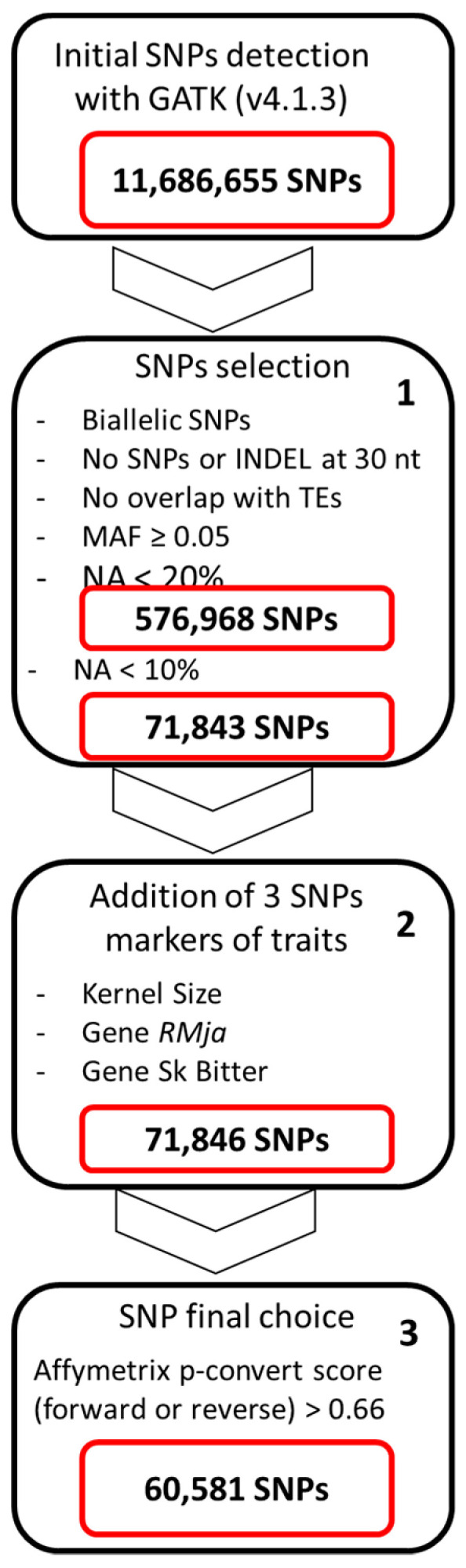
Pipeline of SNP detection, selection and final choice with all filters applied to design the Almond SNP array. (TE: Transposon Element, NA: missing data, MAF: Minor Allele Frequency).

**Figure 2 plants-12-00242-f002:**
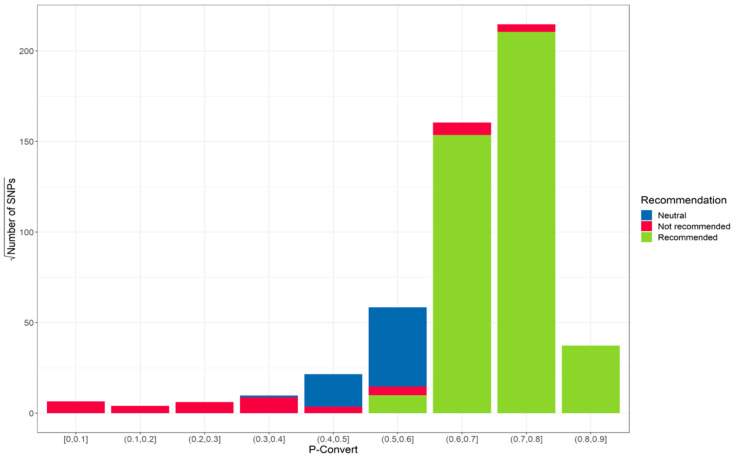
Distribution of the square root of the number of SNPs based on p-convert score and colored by recommendation from Affymetrix in-silico analyses. (P-convert = Probability of conversion based on the best selected strand of the probe sequence.)

**Figure 3 plants-12-00242-f003:**
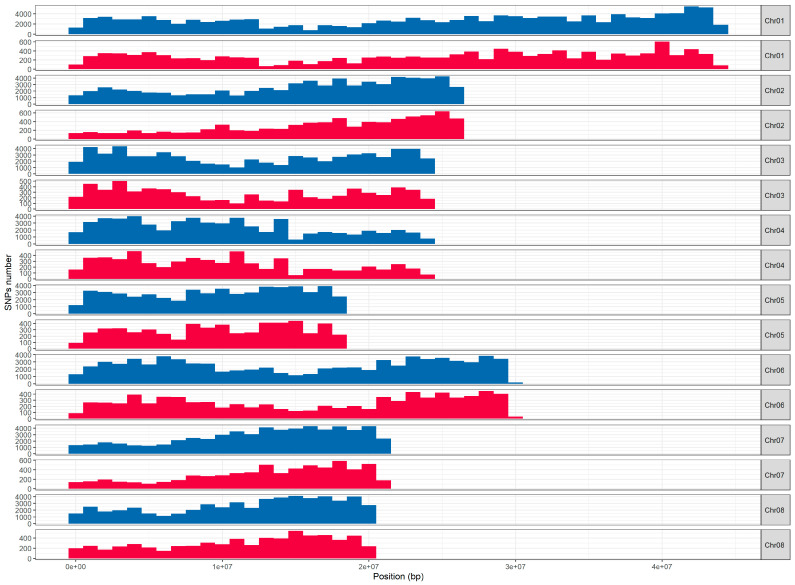
Distribution of SNPs along each chromosome. Each bar represents an interval of 1 Mbp. All the 576,968 SNPs filtered after step 1 are plotted in blue and the 60,581 array SNPs are plotted in red. The abscissa represents the physical position.

**Figure 4 plants-12-00242-f004:**
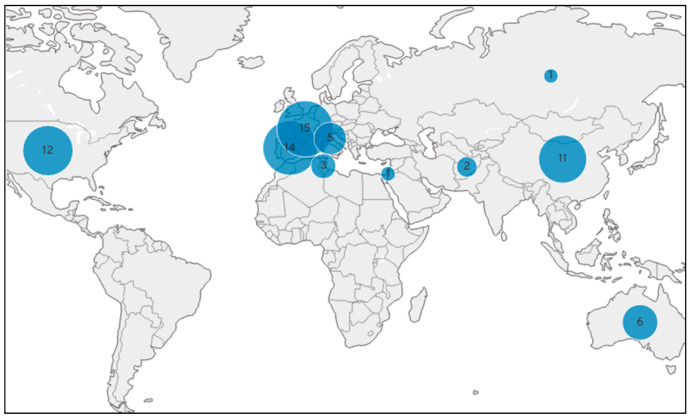
Origins of the resequenced accessions.

**Table 1 plants-12-00242-t001:** Position of the three SNPs linked to different traits.

SNP Name	Chr	Position_V2	Position_V1	SNP_Ref	SNP_Alt	Trait	Gene
AX-599403222	Pd07	9,019,871	7,763,091	A	G	RK Nematode R	*RMja*
AX-599403226	Pd01	3,296,885	2,496,687	T	C	Kernel weight	
AX-599403227	Pd05	11,867,647	11,521,869	C	T	Bitter taste	*Sk*

V1: *Prunus dulcis* Lauranne Genome v1.0. a1; V2: *Prunus dulcis* Texas Genome v2.0.

**Table 2 plants-12-00242-t002:** Repartition of SNPs in the six categories.

SNP Category	Number	Percent
Poly High Resolution (PHR)	47,012	77.6
Mono High Resolution (MHR)	85	0.1
No Minor Hom (NMH)	3074	5.1
Call Rate Below Threshold (CBT)	2435	4
Off-Target Variant (OTV)	3425	5.7
Other	4550	7.5
Total	60,581	

**Table 3 plants-12-00242-t003:** Genotyping results of some *Prunus* species and interspecific hybrid accessions with the almond array. (NA: Missing data, He: Heterozygote).

Species	Accessions	NA Number	NA Percent	He Number	He Percent
*P. dulcis*	185 *	262–1665	0.4–2.7	7077–14,065	11.9–23.4
*P. bucharica*	1	3099	5	3242	6
*P. fenzliana*	9	1448–1937	2.4–3.2	3516–4267	5.9–7.3
*P. kuramica*	1	4212	7	3599	6
*P. dulcis × P. dehiscens*	2	576–1208	1–2	10,126–11,792	17.1–19.7
*P. dulcis × P. webbii*	1	570	1	14,976	25
*P. persica*	2	7103–9412	11.7–15.5	4965–5897	9.3–11.5
*P. davidiana*	2	5610–5671	9.3–9.4	4598–4822	8.4–8.8
*P. persica × P. dulcis*	3	1240–2003	2–3.3	6928–11,277	11.8–19
*P. bucharica × P. persica*	1	2130	4	4643	8
*P. pedunculata*	1	12,806	21	6809	14

* There were 2 other almond accessions with 7.3% and 12.1% of missing data.

**Table 4 plants-12-00242-t004:** Efficient markers to create genetic map for a F2 biparental population. (LG: Linkage group, He = Heterozygote, Nb = Number.)

	SNP Nb on “Penta”		Map	
LG	He	*p*-Value (χ²) ≤ 0.01	Marker Nb	Size (in cM)
1	2055	1541	261	208
2	1339	534	81	69
3	1146	1146	166	108
4	1302	574	96	89
5	1042	451	87	96
6	1456	1402	190	114
7	1176	841	145	99
8	1158	745	133	94
Total	10,674	7234	1159	876

**Table 5 plants-12-00242-t005:** Number of SNPs for each allelic state for the genotyped “Texas” accession.

Allele State *	Number	Percent
0	109	0.2
1	10,416	17.2
2	49,765	82.1
NA	291	0.5
Total	60,581	

* allele common number with the reference “Texas”, NA: missing data.

**Table 6 plants-12-00242-t006:** Error numbers between each replicate of Ferrastar.

Replicate	R2	R3
R1	549 (0.90%)	576 (0.95%)
R2	-	615 (1.01%)

**Table 7 plants-12-00242-t007:** Number of monomorphic SNPs in the F1 Penta and erroneous data points in the 49 hybrids of the F2 progeny.

F1 (“Penta”)		F2 Progeny				Erroneous Data Points	
	Marker Nb	AA	AB	BB	Total	Nb	%
AA	20,462	999,542	494	2167	1,002,203	2661	0.27
BB	22,276	1324	332	1,089,549	1,091,205	1656	0.15

## Data Availability

The probe set sequences and positions of all SNPs included in the almond chip from this study have been deposited in the GDR database under ID code tfGDR1059 and the data will be available for download at https://www.rosaceae.org/publication_datasets (accessed on 28 December 2022). The Axiom 60K almond SNP array (PdSNP_V1) is commercially available with Thermo Fischer Scientific, with request to the author before 2024.

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
