# Peer review of "Development and Evaluation of an AxiomTM 60K SNP Array for Almond (Prunus dulcis)"

_plants, 2023, doi:10.3390/plants12020242_

Round 1
Reviewer 1 Report
The manuscript fits the scope of the Plants journal and should be interesting for its readers. Also, the publication of the submitted text as 'Communication' seems reasonable, given the technical nature of the article.
I find the manuscript well prepared and structured (in terms of general outline, given the provided research. However, I think the manuscript requires fine rewriting to improve its style and grammar, which in some cases may cause misunderstandings. Also, some sections, particularly in the introduction, require attention to improve the paper's clarity and avoid repetition.
Below I enclose some of the remarks, which might be helpful when improving the manuscript before the final acceptance.
The Latin name of the species (or genera) should be included in the title
Line 19, 'Axiom® almond 60K SNP (PdSNP_V1)' should be changed to 'Axiom® 60K almond SNP array (PdSNP_V1).'.
Line 29, it is unclear to what' 0.27% and 0.15%' refers.
Line 46, full Latin names of species or genera of almond and peach should be provided.
Line 48, unclear to what the word 'they' refers.
Line 51, mention precisely Latin name of 'Texas' and 'Lauranne’ varietas.
Line 57, the sentence starting with ‘We developed…’ is somehow in the wrong place, because the next chapter mentions that ‘In this study, we present….’ And actually repeats the same statement. Please reorganize and clarify.
Line 66, the name used for the array, is inconsistent across the manuscript (pdSNPV1, PdSNP_V1, etc), make it consistent throughout the text
Line 110, should be ‘Distribution of Array SNPs’
Line 125, should ‘Array’ be with a large A letter?
Figure 4 description. Remove the word ‘countries’ because it is not possible to infer the country of origin precisely from that figure.
247-259 – check the interline spacing or font size.
295, shouldn’t it be ng/ul ? instead of ul/ng ?
316, change ‘data are reliable’ to ‘genotyping is reliable’
317-318 – change style and rephrase: based on F2 progeny, we found only a few errors indicating departures from Mendelian inheritance of SNP loci
Author contribution; supervision could not be made by all authors, I understand ‘supervision’ as the activity that takes care and controls all essential steps during research and preparation of the article, therefore, this could be done by the senior author, but not all coauthors.
Author Response
Dear Reviewer,
I have modified my manuscript as recommended by you and the other reviewer (new version joined). I have used the “track changes” feature in Microsoft Word when making revisions
Here is the response (in red) to the remarks you made , included in your comments (in black)
I find the manuscript well prepared and structured (in terms of general outline, given the provided research. However, I think the manuscript requires fine rewriting to improve its style and grammar, which in some cases may cause misunderstandings. Also, some sections, particularly in the introduction, require attention to improve the paper's clarity and avoid repetition.
I have rewritten the abstract and modify the introduction; I have also rearranged some parts between the results and the materiel and methods, as recommended by the other reviewer .
My colleague Michelle Wirthensohn from Australia has corrected English language
Below I enclose some of the remarks, which might be helpful when improving the manuscript before the final acceptance.
I have taken your comments into account
The Latin name of the species (or genera) should be included in the title : yes
Line 19, 'Axiom® almond 60K SNP (PdSNP_V1)' should be changed to 'Axiom® 60K almond SNP array (PdSNP_V1).'. I have changed all in the text
Line 29, it is unclear to what' 0.27% and 0.15%' refers. It was the error rate percentage between the expected and observed point for the two groups of homozygote
Line 46, full Latin names of species or genera of almond and peach should be provided. ; I have provided it
Line 48, unclear to what the word 'they' refers. It was the authors, I have modified the sentence
Line 51, mention precisely Latin name of 'Texas' and 'Lauranne’ varietas. : Prunus dulcis has been added
Line 57, the sentence starting with ‘We developed…’ is somehow in the wrong place, because the next chapter mentions that ‘In this study, we present….’ And actually repeats the same statement. Please reorganize and clarify. I have done it
Line 66, the name used for the array, is inconsistent across the manuscript (pdSNPV1, PdSNP_V1, etc), make it consistent throughout the text. I have standardized it
Line 110, should be ‘Distribution of Array SNPs’ done
Line 125, should ‘Array’ be with a large A letter? I changed with “a”
Figure 4 description. Remove the word ‘countries’ because it is not possible to infer the country of origin precisely from that figure. OK , I have changed
247-259 – check the interline spacing or font size. OK
295, shouldn’t it be ng/ul ? instead of ul/ng ? yes it was a mistake
316, change ‘data are reliable’ to ‘genotyping is reliable’ OK
317-318 – change style and rephrase: based on F2 progeny, we found only a few errors indicating departures from Mendelian inheritance of SNP loci OK
Author contribution; supervision could not be made by all authors, I understand ‘supervision’ as the activity that takes care and controls all essential steps during research and preparation of the article, therefore, this could be done by the senior author, but not all coauthors. Effectively , it was mainly me that did the supervision

Reviewer 2 Report
The manuscript (plants-2041375) entitled “Development and evaluation of an AxiomTM 60K SNP array almond”. A high-density, high-efficiency and robust single nucleotide polymorphism (SNP) array are essential to enable faster progress in plant breeding for new cultivar development. In this regard, the authors have made efforts to develop 60K SNP array by resequencing 81 almond accessions. There are a lot of grammatical errors and a lack of continuity. I would strongly suggest the authors revise the entire manuscript and resubmit it. The manuscript cannot be accepted in the present form and the comments of the review should be addressed satisfactorily and modified accordingly. The paper may be considered only after a satisfactory reply and compliance with the observations along with major revision.
Line 24: “NoMinorHomand” delete and “NoMinorHom"
Line 24-32: Not clear, Abstract should be rewritten, there is no continuity and confusion, start with the background of the study, specific objective of your study, what is the purpose of SNP array and final outcome of your study.
Line 36-40: Cite references
Line 56: The set of resequenced varieties was chosen to include a wide diversity but also progenitors of cultivated varieties. Not clear (rewrite)
Line 61-67: Why it is in the introduction part, it should be in the discussion part. Mention your specific objectives and approaches at the end of the introduction
Line 68: In the Results – Discussion part, Material Methods is completely mixed with results and the entire discussion part is discussed without proper citations.
Line 80: Why only 3 SNP were selected??
Line 113-115: The density of SNP is highest along the chromosome arms and lowest in the centromeric regions for both distributions. Explain the reason behind it with citations???
Line 125-128: Accessions, genotypes, hybrids, varieties are mentioned, be specific and use one common term in the manuscript.
Line 130-132: Abrrevate (MHR) (NMH)
Line 245: Materials and Methods (Rewrite)
Author Response
Please see the attachment of a coverletter and a new version of the manuscript

Round 2
Reviewer 2 Report
I found this research article interesting and the authors have revised the manuscript. However minor corrections are to be attended to. Overall this research article contains quite useful information and deserves to be published. I would like to see a clean copy before publishing.
Author Response
Dear reviewer
Thank you for your new comments
My colleague Michelle Wirthensohn from Australia has corrected English language of the new revised manuscript.
this document is intended for communication so I hope that these last corrections will be sufficient.
Best regards